# Controlling of Photophysical Behavior of Rhenium(I) Complexes with 2,6-Di(thiazol-2-yl)pyridine-Based Ligands by Pendant π-Conjugated Aryl Groups

**DOI:** 10.3390/ijms231911019

**Published:** 2022-09-20

**Authors:** Anna M. Maroń, Joanna Palion-Gazda, Agata Szłapa-Kula, Ewa Schab-Balcerzak, Mariola Siwy, Karolina Sulowska, Sebastian Maćkowski, Barbara Machura

**Affiliations:** 1Institute of Chemistry, University of Silesia, Szkolna 9, 40-006 Katowice, Poland; 2Centre of Polymer and Carbon Materials, Polish Academy of Sciences, M. Curie-Sklodowska 34, 41-819 Zabrze, Poland; 3Nanophotonics Group, Institute of Physics, Faculty of Physics, Astronomy and Informatics, Nicolaus Copernicus University, Grudziadzka 5, 87-100 Torun, Poland

**Keywords:** rhenium(I) carbonyls, 2,6-di(thiazol-2-yl)pyridines, aryl fused rings, photoluminescence, ground- and excited-state properties, excited-state equilibrium, femtosecond transient absorption

## Abstract

The structure–property correlations and control of electronic excited states in transition metal complexes (TMCs) are of high significance for TMC-based functional material development. Within these studies, a series of Re(I) carbonyl complexes with aryl-substituted 2,6-di(thiazol-2-yl)pyridines (Ar^n^-dtpy) was synthesized, and their ground- and excited-state properties were investigated. A number of condensed aromatic rings, which function as the linking mode of the aryl substituent, play a fundamental role in controlling photophysics of the resulting [ReCl(CO)_3_(Ar^n^-dtpy-κ^2^N)]. Photoexcitation of [ReCl(CO)_3_(Ar^n^-dtpy-κ^2^N)] with 1-naphthyl-, 2-naphthyl-, 9-phenanthrenyl leads to the population of ^3^MLCT. The lowest triplet state of Re(I) chromophores bearing 9-anthryl, 2-anthryl, 1-pyrenyl groups is ligand localized. The rhenium(I) complex with appended 1-pyrenyl group features long-lived room temperature emission attributed to the equilibrium between ^3^MLCT and ^3^IL/^3^ILCT. The excited-state dynamics in complexes [ReCl(CO)_3_(9-anthryl-dtpy-κ^2^N)] and [ReCl(CO)_3_(2-anthryl-dtpy-κ^2^N)] is strongly dependent on the electronic coupling between anthracene and {ReCl(CO)_3_(dtpy-κ^2^N)}. Less steric hindrance between the chromophores in [ReCl(CO)_3_(2-anthryl-dtpy-κ^2^N)] is responsible for the faster formation of ^3^IL/^3^ILCT and larger contribution of ^3^ILCT_anthracene__→__dtpy_ in relation to the isomeric complex [ReCl(CO)_3_(9-anthryl-dtpy-κ^2^N)]. In agreement with stronger electronic communication between the aryl and Re(I) coordination centre, [ReCl(CO)_3_(2-anthryl-dtpy-κ^2^N)] displays room-temperature emission contributed to by ^3^MLCT and ^3^IL_anthracene_/^3^ILCT_anthracene__→__dtpy_ phosphorescence. The latter presents rarely observed phenomena in luminescent metal complexes.

## 1. Introduction

In 1992, Ford and Rodgers demonstrated a new strategy to prolong the luminescence lifetimes of Ru(II) polypyridine emitters by coupling them with an organic chromophore possessing its lowest-lying and long-lived triplet IL (Intraligand) state close in energy to the triplet MLCT (Metal-to-Ligand Charge Transfer) state of the metal-based chromophore [1]. The origin of this behavior lies in the formation of thermal equilibration between ^3^MLCT and ^3^IL excited states and reversible electronic energy transfer between them. The non-emitting triplet state of the appended hydrocarbon chromophore acts as an excited-state storage element, leading to a prolongation of the MLCT emission. Compounds with sufficiently long luminescence lifetimes can find applications as sensors, providing high-quality time-resolved photoluminescence images. They are also able to transfer energy from the ^3^MLCT/^3^IL excited state to generate reactive oxygen species or singlet oxygen from molecular oxygen [2,3].

Since 1992, many transition metal polypyridine complexes with appended organic chromophores have been widely investigated. Good results have been achieved with the use of π-conjugated aryl groups, such as anthryl, pyrenyl or naphthalimides [4,5,6,7,8,9,10,11,12,13,14]. A striking example is the Re(I)-carbonyl complex bearing N-(1,10-phenanthroline)-4-(1-piperidinyl)naphthalene-1,8-dicarboximide with approx. 3000-fold longer excited-state lifetime relative to the parent complex [ReCl(CO)_3_(phen)] (phen-1,10-phenanthroline) [15]. Most remarkably, even small structural modification of an organic chromophore can induce significant changes in the photophysical behaviour of transition metal complexes [16], as demonstrated for [Re(5-Rphen)(CO)_3_(4-dimethylaminopyridine)](PF_6_) with the naphthalimide unit systematically modified at the 4-position (R).

In the current work, a series of aryl-substituted 2,6-di(thiazol-2-yl)pyridines (Ar^n^-dtpy) were used to synthetize [ReCl(CO)_3_(Ar^n^-dtpy-κ^2^N)] (Figure 1), and the photobehavior of resulting Re(I) carbonyl complexes has been thoroughly investigated by cyclic voltammetry, absorption and emission spectroscopies, transient absorption, in conjunction with density functional theory (DFT) and time-dependent DFT methods.

The 2,6-Di(thiazol-2-yl)pyridines were designed as structural analogues of 2,2′:6′,2′′-terpyridines (terpys), which represent one of the most extensively used organic building blocks in supramolecular and coordination chemistry [17,18,19,20,21,22,23]. Transition metal complexes with 2,2′:6′,2′′-terpyridines and their structural analogues have recently received extensive attention due to their thermal stability, kinetic inertness, rich optical and electrochemical properties relevant for their potential applications as effective luminescent materials and promising anticancer agents [24,25,26,27,28,29]. The photoluminescence and biological features of these systems are generally fine-tuned by two approaches: (i) incorporation of appropriate substituents in the 4′-position of the central pyridine ring of 2,2′:6′,2′′-terpyridine [30,31,32], and (ii) substitution of peripheral pyridine rings of 2,2′:6′,2′′-terpyridine by other heterocycles [33,34,35]. Most importantly, the employment of a newly designed ligand translates to novel excited-state and physical properties of resulting transition metal complexes, and in consequence, new application potential in biology and optoelectronics. The presence of sulphur atoms in the peripheral rings of Ar-dtpy render the dtpy moiety more electron-withdrawing than the terpy itself, which may give rise to stabilization of MLCT states in the resulting transition metal complexes relative to those with terpy-based analogues [36,37,38,39]. Moreover, the thiazole ring is one of the major pharmacophores in many bioactive molecules, and some transition metal complexes with 2,6-bis(thiazol-2-yl)pyridines were successfully demonstrated as cytotoxic agents [40,41,42,43,44]. On the other hand, the attachment of π-extended polycyclic aromatic hydrocarbons into N-donor ligands is one of the most successful strategies in the enhancement of luminescence lifetimes of transition metal complexes [4,5,6,7,8,9,10,11,12,13,14,15,16]. In the context of biological applications, the incorporation of planar aromatic substituents is expected to enhance non-covalent affinities to the DNA of the resulting complexes [45,46,47,48], and prolonged excited-state lifetimes of these systems are beneficial for the generation of reactive oxygen species (ROS) [49,50,51,52,53,54].

The aryl groups attached to the dtpy core in the current work differ among themselves in a number of fused rings and molecular configurations (linear anthracene and nonlinear phenanthrene). In addition, they were introduced in to dtpy via their different positions (1-naphthyl and 2-naphthyl, 9-anthryl and 2-anthryl) leading to different relative orientations of the aryl and {ReCl(CO)_3_(dtpy-κ^2^N)} chromophores. Therefore, the designed Re(I) complexes (Figure 1) offer the opportunity to explore the impact of a number of condensed aromatic rings of the aryl substituent and some spatial effects (the molecular configuration of the appended aryl group and mutual orientation of the aryl and {ReCl(CO)_3_(dtpy-κ^2^N)} chromophores). Within the current work, we demonstrated a pivotal role of the aryl group in controlling photoinduced processes in designed systems [ReCl(CO)_3_(Ar^n^-dtpy-κ^2^N)]. Additionally, to gain a more complete picture of excited-state processes in [ReCl(CO)_3_(Ar-dtpy-κ^2^N)], their ground- and excited-state properties were analysed in comparison to their Re(I) carbonyl analogues bearing 1-naphthyl-, 2-naphthyl-, 9-phenanthrenyl-, and 1-pyrenyl- substituted terpys [55].

The reported findings are useful for understanding and controlling the excited-state nature of Re(I) chromophores, as well as for designing new photoluminescent materials and for various biomedical applications. A growing number of studies have recently highlighted the potential of *fac*-[ReX(CO)_3_(N∩N)]^0/1+^ complexes as chemotherapeutic drugs. Some of the examined Re(I) systems reached or even exceeded the anti-proliferative activities of cisplatin on various human cancer cell lines [56,57,58,59,60,61,62,63].

## 2. Results and Discussion

### 2.1. Synthesis and Characterization

The complexes [ReCl(CO)_3_(Ar^n^-dtpy-κ^2^N)] (**1**–**6**) were prepared using the standard procedure based on the reaction of [Re(CO)_5_Cl] with one equivalent of the corresponding ligand Ar^n^-dtpy. All syntheses resulted in satisfactory yield, and the complexes **1**–**6** were identified via ^1^H and ^13^C NMR spectroscopies, FT-IR technique, and elemental analysis (Appendix A). Due to the bidentate coordination mode of Ar^n^-dtpy to the rhenium(I) centre, the protons of the peripheral thiazolyl rings become magnetically inequivalent, giving separated signals in the ^1^H NMR spectra of **1**–**6** (Appendix A). The singlet assigned to the chemically and magnetically equivalent protons of the central pyridine of Ar^n^-dtpy was found to be affected by the appended aryl group. For **1** (8.72 ppm), **3** (8.79 ppm), **4** (8.79 ppm), and **6** (8.79 ppm) compounds, the considered signal is shifted upfield compared to **2** (8.95 ppm), and **5** (8.99 ppm). The facial arrangement of CO groups in [ReCl(CO)_3_(Ar^n^-dtpy-κ^2^N)] was indicated by a typical pattern of ν(C≡O) absorptions, including a sharp and intense high-energy carbonyl stretching band (2016–2023 cm^–1^) accompanied by two overlapping lower-energy ν(C≡O) absorptions (1937–1880 cm^–1^). Depending on the appended aryl group, there are marginal variations in the average CO stretching frequencies (Appendix A).

In addition, molecular structures of **1**–**4** were confirmed by X-ray analysis (Appendix A in ESI). As shown in Figure 1, the rhenium(I) ion is coordinated to three carbonyl groups in *fac*-arrangement, chloride ion and two nitrogen atoms of Ar^n^- dtpy ligand. Due to κ^2^N-coordination of the dtpy-based ligand and strong steric interaction of the uncoordinated thiazolyl ring with the carbonyl group C(2)–O(2), the octahedral geometry around the Re(I) ion in **1**–**4** is strongly distorted. The distortion of [ReCl(CO)_3_(Ar^n^-dtpy-κ^2^N)] molecules is manifested mostly by a small N(2)–Re(1)–N(1) bite angle [74.24(12)–74.7(3)°], noticeable enlargement of C(2)–Re(1)–N(2) angle [99.4 (3)–102.22(19)°], and elongation of Re–N_pyridine_ [2.234(4)–2.241(3) Å] compared to Re(1)–N_thiazole_ [2.134(7)–2.160(3)Å]. Comparing the bond distances and angles around the Re(I) ion in the examined complexes, only minor differences are observed (Appendix A). Noticeable differences amongst examined structures seen as the twisting of the pendant aryl group relative to the central pyridine ring of dtpy core are taken into consideration. The replacement of 1-naphtyl (**1**) and 9-anthryl (**3**) by appropriate 2-naphtyl (**2**) and 2-anthryl (**4**) induces a decrease in the twist angle. While the pendant substituent of **2** [14.64°] and **4** [8.03°] remains near co-planar with the central pyridine plane, the dihedral angle between the central pyridine and appended group is 57.70° for **1** and 86.64° for **3**. The crystal packing of **1**–**4** is largely dominated by π•••π stacking interactions involving the 2,6-di(pyrazin-2-yl)pyridine core and aryl groups, and additionally stabilized by weak intra- and inter-molecular C–H•••X (X = Cl, S or O) contacts (Figure 2; Appendix A in ESI).

The DSC investigation showed that Re(I) complexes **1**–**6** obtained after synthesis as crystalline compounds can be converted into an amorphous state except for molecule **6**, which melted with decomposition. They showed high melting (T_m_) and glass transition (T_g_) temperatures ranging from 155–303 °C and 123–184 °C, respectively (cf. Experimental part Section Synthesis). The effect of the number of condensed aromatic rings and linking mode of the aryl substituent on the thermal behavior of [ReCl(CO)_3_(Ar^n^-dtpy-κ^2^N)] was observed. In the cases of **2** and **4,** the crystal-to-crystal phase transition (T_c-c_), which may be attributed to the change of packing arrangements and intermolecular interactions of the polymorphs, was observed (see Appendix A). Comparing Re(I) complexes with three fused-ring substituents (**3**–**5**), it can be noticed that the Re(I) complex with anthracene attached at its **2**-position (**4**) exhibited higher T_m_ but lower T_g_ with respect to **3** and **5**. Moreover, **3** and **5** form stable molecular glasses contrary to **4**, for which melting upon heating above T_g_ was confirmed.

### 2.2. Redox Properties

The redox properties of Re(I) complexes **1**–**6** were examined in CH_2_Cl_2_ solution by a combination of cyclic voltammetry (CV) and differential pulse voltammetry (DPV) on a glassy carbon-working electrode and with reference to the redox couple of ferrocene/ferrocenium. All investigated Re(I) complexes exhibited multi-stage oxidation and reduction processes (Appendix A). The values of $E_{1red}^{onset}$ and $E_{1ox}^{onset}$, which were used to estimate the ionization potential (IP), electron affinity (EA), and electrochemical band gap (E_g_), are collected in Table 1. With reference to [64], IP and EA can be correlated with HOMO and LUMO energy levels, respectively.

The onset potentials of the first quasi-reversible reduction processes for complexes **1**–**6** are very close to each other and fall in the range observed for related Re(I) tricarbonyls with 2,6-di(thiazol-2-yl)pyridine-based ligands. On the basis of our earlier works [36,37,38,39], the first reduction wave in 1–6 can be safely assigned to the reduction localized on the dtpy core. The LUMO of [ReCl(CO)_3_(R-dtpy-κ^2^N)] resides on the pyridine and thiazole rings coordinated to the Re(I) ion, and its energy is unaffected by the substituent variations. Compared to the Re(I) carbonyl analogues bearing aryl-substituted terpys [55], the complexes [ReCl(CO)_3_(Ar-dtpy-κ^2^N)] are easier to reduce. The reduction onset potentials of **1**–**6** occur in a more positive range (from −1.52 V to −1.48 V) in relation to those of [ReCl(CO)_3_(Ar-terpy-κ^2^N)] (from −1.67 V to −1.63 V), supporting the assumption that the introduction of sulphur atoms into the peripheral rings of dtpy lowers the LUMO energy of **1**–**6** with respect to that of the corresponding [ReCl(CO)_3_(Ar-terpy-κ^2^N)] one.

The onset potentials of the first irreversible oxidation processes in **1**–**6** also seem to be insensitive to the variations of π-extended aryl substituents, falling in a narrow range 0.73–0.76 V. By comparing the $E_{1ox}^{onset}$ of **1**–**6** with oxidation onset potentials of the free ligands, however, a striking difference can be noticed between Re(I) complexes with naphthyl (**1** and **2**) and phenanthrenyl (**5**) groups and those bearing anthryl (**3**, **4**) and pyrenyl (**6**) substituents (Appendix A). The first oxidation waves of **1**, **2** and **5** occur at significantly lower potentials in relation to the free ligand, so they can be safely assigned to the metal-centred Re(I/II) oxidation process. Conversely, the $E_{1ox}^{onset}$ values of **3**, **4** and **6** are comparable to those of corresponding 2,6-di(thiazol-2-yl)pyridines substituted with anthryl and pyrenyl groups [34]. It is probable that the first oxidation wave of complexes **3**, **4** and **6** corresponds to the oxidation of the pendant aryl unit to its radical cation. Nevertheless, involvement of the Re(I)-based oxidation process cannot be excluded for these systems, as previously reported for the related system [Ru(X-tpy)(bpy-COOH)(NCS)](PF_6_) (where X-terpy = 4′-(9-anthryl)-2,2′:6′,2″-terpyridine and bpy-COOH = 4,4′-dicarboxy-2,2′-bipyridine) [65]. The electrochemical band gap values correlate well with optical ones, which also show small variations depending on the appended π-extended aryl group.

### 2.3. Absorption Spectral Features–Experimental and Theoretical Insight

The electronic absorption spectra of **1**–**6** are shown in Figure 3, Appendix A (in ESI), and summarized in Appendix A. The compounds **1**, **2** and **5** feature intense bands below 350 nm due to π→ π* transitions localized on the organic ligand and much weaker and broad absorption in the range 350–550 nm attributed to the electronic transitions of MLCT character (Figure 3a). The corresponding free ligands Ar^n^-dtpy and aromatic hydrocarbons (naphthalene and phenanthrene) have no intense absorptions at wavelengths longer than 300 nm (see Appendix A). The MLCT positions of **1**, **2** and **5** are almost unaffected by the appended aryl group. The differences between these compounds are noticeable when molar absorption coefficients of the MLCT band are concerned. A significant intensity increase in visible light absorption for **5** is attributed to the introduction of a more extended phenanthrenyl group into the dtpy core, while a slight absorptivity increase of **2** in relation to **1** can be rationalized by the stronger electronic coupling between the 2-nathyl and dtpy units than 1-nathyl and dtpy ones. As a result of the replacement of terpy core by dtpy, the lowest energy band of **1**, **2** and **5** moves towards the red end of the spectrum, in agreement with stabilization of the LUMO due to introduction of sulphur atoms into peripheral rings.

For compound **3**, bearing the 9-anthryl-dtpy ligand, the characteristic vibronic progression for the anthracene (An) appears superimposed on the MLCT band, consistent with the limited electronic coupling between the 9-anthryl and dtpy units due to the conformation restriction exerted by the steric hindrance of the rotation about the C−C linker. Conversely, the compound **4**, with 2-anthryl-dtpy ligand, shows lack of the vibronic progression for the electronic transitions of anthracene (379 nm, 360 nm, 340 nm, 326 nm) and enhanced red-shifted and structureless charge-transfer (CT) absorption in the range 400–550 nm (with maximum at 438 nm) [55,66]. These observations are indicative of strong electronic coupling between the 2-anthryl and dtpy moieties due to the reduced steric hindrance of the rotation about the C−C bond. Such findings correlate well with the X-ray analysis, which showed that the replacement of 9-anthryl (**3**) with 2-anthryl (**4**) leads to a decrease in the dihedral angle between the pendant aryl group relative to the central pyridine ring of dtpy, 86.64° for **3** and 8.03° for **4**. The compounds **3** and **4** represent examples that confirm the photophysical properties of bichromophoric systems can be controlled by the relative orientation of anthracene and {ReCl(CO)_3_(dtpy-κ2N)} chromophores. Moreover, for compound **6**, with the 1-pyrenyl-dtpy ligand, the characteristic pyrene vibronic progression between 290 nm and 350 nm is not observed in its UV–Vis spectrum (Appendix A). Similar to **4**, the complex **6** exhibits broad and enhanced CT absorption with maxima at 427 nm in CHCl_3_ and at 406 nm in MeCN, attributed to the red-shift of the intramolecular charge transfer band of 1-pyrenyl-dtpy superimposed on the MLCT absorption of [ReCl(CO)_3_(4-(1-pyrenyl-dtpy-κ^2^N)] (**6**). Compared to **4**, the CT absorption of **6** is less pronounced indicating the occurrence of weaker electronic coupling between the 1-pyrenyl and dtpy moieties.

For all compounds, the longest wavelength absorption band in chloroform is bathochromically shifted by 25 nm for **1**, 27 nm for **2**, 14 nm for **3** and **4**, 20 nm for **5**, and 19 nm for **6** with reference to that in more polar acetonitrile solution (see Appendix A), representing negative solvatochromism typical for MLCT absorption bands of rhenium(I) tricarbonyl diimine complexes [67,68]. Considering the presented results in film (Figure 3d and Appendix A), it can be seen that the UV–Vis properties of designed Re(I) compounds are similar in solution and in film.

The band attributions are supported by the results of the quantum mechanical calculations (see ESI: Appendix A summarizing the results of DFT and TDDFT calculations). The calculated electronic transitions are overlaid with their experimental spectra (Figure 4 and Appendix A). For compounds **1**, **2** and **5**, three or four energetically lowest singlet transitions, HOMO/H-1/H-3→ LUMO (**1**), HOMO/H-1/H-3→ LUMO/L+1 (**2**), and HOMO/H-1/H-2/H-3/H-3→ LUMO (**5**) were assigned to the absorption in the visible region. All are of the MLCT nature and are characterized by low oscillator strengths due to poor overlap of the molecular orbitals involved in the electronic excitations. The LUMO of **1**, **2** and **5** corresponds predominately to π* orbitals of the pyridine and thiazole rings coordinated to the Re(I) ion, while the above given HOMOs are predominantly Re(I) d*_π_*orbitals with admixture with π*_CO_ and π_Cl_ orbitals (see Figure 4 and Appendix A). The changes in HOMO–LUMO gaps for **1**, **2** and **5** are rather negligible: 3.45 eV for **1**, 3.50 eV for **2** and 3.52 eV for **5** (see Appendix A presenting molecular orbital diagram for studied complexes), so the MLCT absorption of **1**, **2** and **5** falls in the same range. Conversely, following the TD-DFT calculations, the lowest energy absorption bands of **3**, **4** and **6** are attributed to a combination of ^1^MLCT and ^1^ILCT/^1^IL transitions. For these compounds, the transition HOMO → LUMO is of ^1^ILCT/^1^IL nature. The HOMO resides predominately on the aryl unit, while the LUMO is located on the aryl and dtpy moieties. ^1^ILCT transitions originate from charge delocalization from the aryl group to dtpy acceptor moiety. The orbitals 5d_π_ rhenium, π*_CO_ and π_Cl_ have noticeable contribution in the lower energy occupied MOs (H-1 to H-3), and the transitions H-1→ L and H-2→ L possess the MLCT character. The HOMO of [ReCl(CO)_3_(Ar-dtpy-κ^2^N)] with anthryl and pyrenyl substituents (**3**, **4** and **6**) is noticeably destabilized relative to the HOMO levels of **1**, **2** and **5**.

The raised HOMO energy level of **3**, **4** and **6** leads to a decrease in the HOMO–LUMO gap in these systems: **3** (3.23 eV), **4** (3.19 eV) and **6** (3.34 eV), which is manifested in the bathochromic shift of the lowest energy absorption bands of **3**, **4** and **6** as compared to that of **1**, **2** and **5**. Higher oscillator strength values for the calculated low-energy excitations of **4** in relation to those of **3** well correlate with a noticeable increase in molar absorption coefficients of the low-energy absorption bands of **4**, and may be explained by the reduced steric hindrance and stronger coupling of Ar and dtpy moieties for 2-anthryl-dtpy (Figure 3b and Figure 4).

### 2.4. Luminescence Studies

Prior to photoluminescence (PL) studies of **1**–**6** in solution at room temperature (RT), their stability and photostability were investigated (Appendix A). Regarding the partial dissociation of Ar-dtpy ligand from [ReCl(CO)_3_(Ar-dtpy-κ^2^N)] dissolved in polar solvents: acetonitrile and DMF (Appendix A), photoluminescence properties of Re(I) complexes were examined at room temperature only in deaerated CHCl_3_. Additionally, PL studies were performed in a rigid matrix at 77 K (MeOH:EtOH, 1:4), as well as in the solid state as a powder and thin film on a glass substrate and as a blend with poly(N-vinylcarbazole) (PVK) (50 wt. %): 2-(4-tert-butylphenyl)-5-(4-biphenylyl)-1,3,4-oxadiazole. The emission spectral data of **1**–**6** are summarized in Table 2. The normalized emission spectra of the synthesized Re(I) complexes are shown in Figure 5, Figure 6, Figure 7, Figure 8 and Figure 9 and Appendix A in ESI. The photoluminescence behaviours of **1**–**6** were found to be strongly dependent on both the appended aryl group and the excitation wavelength. In CHCl_3_ solution at RT upon photoexcitation at the low-energy end of the MLCT absorption band (463–500 nm), the complexes **1**, **2** and **5** exhibit broad, unstructured emission centred at 728–731 nm, with quantum yield (0.8%–4.3%) and lifetimes (6.3–6.7 ns), which is typical for a weak luminescence from the ^3^MLCT excited state, as reported previously for [ReCl(CO)_3_(L^n^-κ^2^N)] with 2,6-di(thiazol-2-yl)pyridines substituted with heterocycle groups [39].

When complexes **1**, **2** and **5** in CHCl_3_ solutions are excited at absorption maximum wavelength (~415 nm) at room temperature, the emission spectra are composed of two well-separated bands (Figure 5). The possibility that the emission peak at shorter wavelength is the fluorescence of the free ligand that has dissociated from the complex was precluded as all investigated complexes are stable in chloroform solutions (Appendix A). Furthermore, the higher energy bands of **1**, **2** and **5** are red-shifted in relation to the emission band of the appropriate free ligand reflecting some perturbation from ^1^MLCT (Appendix A). These spectral features may indicate that both ^1^Ar-dtpy and ^1^MLCT excited states are promoted upon irradiation of **1**, **2** and **5** at absorption maximum wavelength at room temperature. Additionally, the energy transfer from the singlet excited state localized on the ligand to ^1^MLCT excited state is incomplete, manifesting in the presence of residual fluorescence [16,69,70,71,72] (Appendix A).

Upon moving from the solution at room temperature to the rigid matrix at 77 K, the emissions of **1**, **2** and **5** move toward higher energies (Appendix A), and lifetimes become significantly longer, which is typical of an emission originating from the ^3^MLCT excited state [73].The almost non-structured profiles of **1**, **2** and **5** frozen-state emission bands confirm that the emitting states at 77 K are still predominantly of the ^3^MLCT nature.

By comparing the results obtained for **1**, **2** and **5** with those previously reported for [ReCl(CO)_3_(4´-Ar-terpy-κ^2^N)] analogues [55], the emission of the former complexes is bathochromically shifted, consistent with the stabilization of the LUMO energy level of [ReCl(CO)_3_(4-Ar^n^-dtpy-κ^2^N)] in relation to [ReCl(CO)_3_(4´-Ar^n^-terpy-κ^2^N)]. Very weak vibronic progression in the frozen-state emission bands of **1**, **2** and **5** demonstrates that the replacement of the terpy core with the dtpy one induces an increase in the separation between ^3^MLCT and ^3^IL_dtpy_ (Appendix A).

For complex **3** in CHCl_3_ at RT, mainly unquenched fluorescence originating from ^1^IL_An_/^1^ILCT_An__→__dtpy_ is detected (Figure 6). Compared to the free ligand, it appears at longer wavelengths, indicating the perturbation from the Re(I) coordination sphere (Appendix A).

The phosphorescence of ^3^MLCT origin in **3** is most probably quenched by the lower-lying triplet state localized on the anthracene moiety, as reported previously for related bichromophoric systems incorporating the anthracene unit [74].

The emission band of **4** in CHCl_3_ excited at the maximum (420 nm) is very broad. By comparing it with the photoluminescence behavior of a free ligand and ^3^MLCT emission band of **1**, **2** and **5**, it can be assumed that the observed photoluminescence is attributed to non-quenched ^1^IL/^1^ILCT fluorescence and ^3^MLCT phosphorescence, but the participation of ^3^IL/^3^ILCT also cannot be excluded. Upon photoexcitation of **4** at the low-energy end of the MLCT absorption band (475 nm), the contribution of the ligand-based phosphorescence becomes more noticeable. It is represented by the structured emission band at longer wavelengths (>680 nm). Remarkably, anthracene-related room-temperature triplet emission has rarely been reported, and its observation has been rationalized by enhanced heavy atom impact [75].

The frozen-state emissions of **3** and **4** become completely different with respect to those of **1**, **2** and **5** at 77 K (Figure 6 and Appendix A). It appears at lower energy, shows well-resolved vibronic structure, and the excited-state lifetimes fall in milliseconds, being two orders of magnitude longer than those of **1**, **2** and **5** at 77 K. By comparison with phosphorescence of the free ligands and organic chromophore (anthracene), it can be assumed that the emissions of **3** and **4** at 77 K occur predominately from the excited state of ^3^IL_An_/^3^ILCT_An__→__dtpy_ character, with a small admixture ^3^MLCT (Appendix A). As clearly evidenced by time-resolved emission spectra (TRES) recorded at 77 K (Figure 7), ^3^MLCT excited state undergoes triplet−triplet energy transfer into ^3^IL_An_/^3^ILCT_An__→__dtpy_. The contribution of ^3^MLCT excited state is noticeably higher in the case of the complex with 9-anthryl appended group (**3**). The complexes **3** and **4** differ also in the relative contribution of residual fluorescence, which is also greater in the case of the complex with weaker electronic coupling of the anthracene and {ReCl(CO)_3_(dtpy-κ^2^N)} chromophores (**3**).

The emission spectrum of the deaerated CHCl_3_ solution of **6** consists of two well-separated bands (Figure 8). The lower-energy emission peak is quenched upon exposure to air, signalling that it arises from a triplet state (Appendix A). Most remarkably, the lower-energy component occurs in the range of both ^3^MLCT and organic chromophore phosphorescence ^3^IL_pyrene_/^3^ILCT_pyrene__→__dtpy_. Contrary to the well-structured frozen-state emission, the RT phosphorescence of **6** is structureless, which is expected for ^3^MLCT.

Notably, the room temperature lifetime of **6** in CHCl_3_ falls in the microsecond range, showing significant elongation relative to the complexes **1**, **2** and **5** with lifetimes from 6.3 ns to 6.7 ns. Such a large lengthening of the excited-state lifetime may be indicative of thermal activation between the closely lying ^3^MLCT and ^3^IL/^3^ILCT.

The phosphorescence energies and character of the lowest energy triplet state of **1**–**6** were also investigated theoretically. The triplet emission energies were determined as the difference between the ground singlet and the triplet state ΔE_T1-S0_, and the character of the lowest energy triplet excited state was determined on the basis of the spin density surfaces generated from the lowest energy triplet state (Figure 9). For complexes **1**, **2** and **5**, the spin density surfaces are distributed on the {Re(CO)_3_Cl} unit and π^*^ orbitals of the pyridine and thiazole rings coordinated to the Re(I) centre, supporting the MLCT character of the triplet emission. The spin density surface plots generated from the T_1_ states of complexes **3**, **4** and **6** demonstrate that the spin density is localized on the aryl substituent and central pyridine of dtpy, corroborating the ^3^IL/^3^ILCT origin of their emission.

Finally, three complexes (**1**, **2** and **5**) which emitted light as thin films (Appendix A) were further investigated in a PVK:PBD matrix with content **2** and 15%wt Re(I) complex. An intense band originating from the matrix emission was observed at higher energy (~400 nm) in addition to the band ascribed to the PL of the Re(I) complexes (Appendix A). Thus, it can be concluded that the energy transfer from the matrix to the luminophore was very weak. The compounds **1**, **2** and **5** were also selected for examination of their abilities, namely emission of light induced by applied voltage. Two types of diodes were prepared containing a neat complex (ITO/PEDOT:PSS/complex/Al) and blend with PVK:PBD (ITO/PEDOT:PSS/PVK:PBD:complex(1/2/15%wt)/Al) as an active layer. In such preliminary investigations, only electroluminescence (EL) spectra were registered considering the effect of various voltages applied on the EL intensity. All tested devices emitted red light with similar maximum of EL band (λ_EL_) about 635–660 nm (cf. Figure 10).

The diodes started emitting light under a rather high voltage of about 9 V, and as the voltage increased, the emission intensity also increased. Considering the results, the effect of pendant aryl structure linked with the dtpy core also significantly impacts the EL ability of the synthesized Re(I) complexes. It was observed that the presence of naphthyl (**1**, **2**) and phenanthrene (**5**) unit promotes the EL properties of complexes compared to other substituents, such as anthracene (**3**, **4**) and pyrene (**6**).

### 2.5. Ultrafast Photodynamics of Representative Complexes ***1***, ***3***, ***4*** and ***6***

To better understand the role of aryl pendant groups in controlling the photophysics of rhenium(I) compounds [ReCl(CO)_3_(Ar^n^-dtpy-κ^2^N)], the excited-state processes in the representative complexes of **1**, **3**, **4**, **6** dissolved in CHCl_3_ were investigated via femtosecond TA spectroscopy upon excitation at 355 nm, with 0.17–0.26 μJ energy per each pulse. The experimental conditions were optimized on the basis of the photodamage and fluence dependence tests in order to prevent any photoproduct formation and to avoid two-photon processes (Appendix A). TA spectra of complexes of **1**, **3**, **4**, **6** along with the results of the global fit analysis are gathered in Figure 11 and Table 3.

Table 1 shows only positive signals across the entire wavelength range (Figure 11). They are generally composed of a broad excited-state absorption (ESA) band in the range 450–675 nm, and noticeably more intense and narrower higher-energy absorption with a clear maximum at 397 nm. While the formation of the lower-energy absorption (450–675 nm) is complete just after photoexcitation and remains almost constant up to 20 ps, the band at 397 nm increases up to hundreds of ps, and then starts to decay. Such spectral features are typical of T_1_→ T_n_ absorptions of the ^3^MLCT excited state in rhenium(I) carbonyls [ReCl(CO)_3_(N^∩^N)] [16,71,76,77]. Based on the findings of Chergui et al. [76,77], the weaker excited-state absorption is assigned to Cl/(N^∩^N) ^•–^→Re (Ligand-to-Metal-Charge-Transfer, LMCT) transitions, while the ESA band in the range 375–450 nm represents transitions of predominant π→π* (IL) character.

Compared to the previously reported [ReCl(CO)_3_(4′-Ar^1^-terpy-κ^2^N)] analogue, the ESA bands 397 nm and 450–675 nm of **1** are red-shifted by 19 nm and 105 nm, respectively [55]. A similar trend, namely bathochromic shift, was also evidenced in steady-state absorption and emission spectroscopic studies of **1** relative to [ReCl(CO)_3_(4′-Ar^1^-terpy-κ^2^N)] and can be assigned to the increase in π-acceptor abilities of Ar^1^-dtpy due to the presence of sulphur atoms in peripheral rings of the organic ligand.

The global fit analysis of fsTA data of **1** revealed three excited-state lifetimes 2.24 ps, 77.2 ps, 6258 ps, represented by decay-associated spectra shown in Figure 11 (top row). The ultrafast ISC process was not observed, as it occurs within a time shorter than the internal response function (IRF = 170 fs) of the experimental setup. All of these findings confirm the same deactivation pathway model of **1** as postulated previously for [ReX(CO)_3_(N^∩^N)]^0/+1^ (N^∩^N = bpy, phen, 4,7-dimethyl-phen, X = Cl or imidazole) [76,77,78], [ReCl(CO)_3_(terpy-κ^2^N)] [79], and [ReCl(CO)_3_(4′-Ar^1^-terpy-κ^2^N)] [55]. The ultrafast ISC with a time constant below IRF of the fsTA experiment leads to the population of two vibronically hot triplet states, intermediate ^3^IL_Ar-dtpy_ and the lower in energy ^3^MLCT. The t_2_ component corresponds to the formation of the intermediate triplet via ISC crossing, its vibrational relaxation and equilibration with the ^3^MLCT state. The pre-last component t_3_ (77.2 ps) is assigned to the later stages of vibrational relaxation and reorganization within a supramolecular system comprising the excited [ReCl(CO)_3_(4′-Ar^1^-terpy-κ^2^N)] chromophore and solvent molecules. Its value is strongly dependent on the solvent and organic ligands. For **1**, changing the solvent from chloroform to a more viscous glyceryl triacetate leads to the increase of this component by an order of magnitude (Appendix A). The component t_4_ represents the recovery of **1** to the ground state, and its value correlates well with the steady-state emission lifetime.

The TA spectra of **3** at early time-delays (up to 2 ps) show a strong resemblance to spectral features of **1** corresponding to T_1_→ T_n_ absorptions of the ^3^MLCT excited state. Following pump–probe delay (5 ps), the initially formed ESA bands with maxima at 398 nm and 567 nm gradually demonstrate a drop in their intensities and undergo conversion into the ESA with a maximum at 431 nm and noticeable shoulder of lower intensity at approximately 510 nm. These spectral features are typical of T_1_→T_n_ transitions of anthracene (An) [80,81]. The signal persists up to the end of the delay stage, as ^3^An lifetime significantly exceeds the maximum pump−probe delay of the fsTA setup [82]. By global fitting, the ultrafast transient data of **3** were resolved into five components with lifetimes of 0.3 ps, 3.07 ps, 94.1 ps, 1800 ps, and infinite residual. The two fastest processes occurring within 0.3 ps and 3.07 ps are associated with the formation of ^3^MLCT via two channels ^1^MLCT → ^3^MLCT and S_1_(An) → ^1^MLCT → ^3^MLCT, respectively. This postulation is well supported by the evolution-associated spectra (EAS) (Appendix A). Both EAS_1_ and EAS_1_ show spectral profiles typical for ^3^MLCT excited state in the Re(I) carbonyl chromophores. The component DAS_3_ with the lifetime t_3_, which is negative in the region corresponding to the lowest triplet state ^3^IL_An_, can be associated with the formation of the triplet state localized on the anthracene in the process S_1_(An) → T_2_(An) → T_1_(An). The S_1_ state of the anthracene is isoenergetic to T_2_(An), and anthracene is one of the rare organic compounds, which show intersystem crossing (ISC) via an upper excited state [83,84]. The decay-associated spectrum DAS_4_ with lifetime ~1800 ps represents intramolecular triplet−triplet energy transfer (TTET) from ^3^MLCT to T_1_(An) excited state, along with vibrational relaxation of the formed T_1_(An) [85]. The fully relaxed excited-state T_1_(An) recovers to the ground state, which is represented by DAS_5_ with infinite lifetime. The photophysical processes occurring upon photoexcitation of **3** are summarized in Figure 2.

The TA spectra of **4** are composed of three ESA bands with maxima at 394 nm, 487 nm, and 639 nm. All these signals appear almost immediately after photoexcitation at 355 nm (at 0.2 ps), and remain visible at the longest delay time experimentally available (7.6 ns). They show a rise in intensity up to 10 ps, and then start to slowly decay. The band at 487 nm can be assigned to T_1_→T_n_ absorptions of the ^3^IL_An_ excited state. Compared to the ESA of ^3^An in **3**, it is bathochromically shifted, which correlates well with steady-state absorption spectroscopy results. In contrast to **3**, the triplet state of **4** also shows a noticeable ^3^ILCT_(An__→__dtpy)_ character. The transient absorption bands at 639 nm and 394 nm can be assigned to the anthracene radical cation and the dtpy radical anion, respectively.

The global fit analysis of fsTA data of **4** revealed three excited-state lifetimes 0.38 ps, 208.8 ps, and infinitive residual (Figure 11). The component with the ultrafast time constant represents the formation of ^3^IL_An_/^3^ILCT_(An__→__dtpy)_ excited state, most likely in the following sequence of processes S_1_(An)/(An→dtpy) → ^1^MLCT → ^3^MLCT → T_1_(An)/(An→dtpy). The population of the triplet excited state in **4** is significantly faster than that in the isomeric complex **3**, which can be rationalized by stronger electronic coupling between the more planar 2-anthryl-dtpy and {ReCl(CO)_3_(dtpy-κ^2^N)} chromophores. The component DAS_2_ (t_2_ = 208.8 ps) is assigned to vibrational relaxation and reorganization within a supramolecular system composed of the excited [ReCl(CO)_3_(4′-Ar^4^-dtpy-κ^2^N)] chromophore and solvent molecules, while DAS_3_ with infinite lifetime corresponds to the ground-state recovery.

Careful examination of fsTA data of **6** allows us to notice a striking difference between the TA spectra up 1 ps and those at longer time delays (see Appendix A). While the first one displays a minimum corresponding to the stimulated emission (SE) 432 nm and ESA band in the range 475–680 nm, TA spectra at 2 ps time delay show a ground-state bleaching (GSB) at 422 nm, which reflects well the shape of the ground-state absorption band of **6**, along with the ESA band with two discernible maxima at 529 nm and 590 nm. By comparing these results with previously reported femto- and nanosecond transient absorption data for free ligand Ar^6^-dtpy [86], the observed spectral features of **6** at early and further time delays can be safely assigned to ^1^IL_pyrene_/^1^ILCT_pyrene__→__dtpy_ and ^3^IL_pyrene_/^3^ILCT_pyrene__→__dtpy_, respectively.

By global fitting, the ultrafast transient data of **6** were resolved into four components with lifetimes 0.5 ps, 7.93 ps, 164 ps, and infinite residual. On this basis, we can postulate that the initially populated ^1^IL_pyrene_/^1^ILCT_pyrene__→__dtpy_ excited state undergoes energy transfer into the ^1^MLCT state via the FRET mechanism, which is followed by ISC crossing and formation of “hot” triplet excited-state ^3^MLCT (t_1_ = 0.5 ps). After its relaxation (t_1_ = 7.93 ps), the triplet−triplet back-energy transfer from the triplet excited state localized on {ReCl(CO)_3_(dtpy-κ^2^N)} framework to the T_1_ state of ^3^IL_pyrene_/^3^ILCT_pyrene__→__dtpy_ occurs (t_3_ = 164 ps). DAS_4_ corresponds to the recovery of the fully relaxed lowest triplet state ^3^IL_pyrene_/^3^ILCT_pyrene__→__dtpy_ to the ground state, as shown in Figure 3.

## 3. Materials and Methods

Commercially available pentacarbonyl chlororhenium(I), Re(CO)_5_Cl (Sigma Aldrich), poly(9-vinylcarbazole) (PVK, M_n_ = 25,000–50,000; Sigma Aldrich), poly(3,4-(ethylenedioxy)thiophene): poly-(styrenesulfonate) (PEDOT:PSS, 0.1–1.0 S/cm, Sigma Aldrich), substrates with pixilated ITO anodes (Ossila) and solvents for synthesis (of reagent grade) and for spectroscopic studies (of spectroscopic grade) were used as received without further purification. All aryl-substituted 2,6-di(thiazol-2-yl)pyridine (**Ar^1^-dtpy** – **Ar^6^-dtpy**) were prepared according to the method reported in our previous work [34].

### 3.1. Synthesis of [ReCl(CO)_3_(Ar^n^-dtpy-κ^2^N)] (***1***–***6***)

The equimolar mixture of precursor [Re(CO)_5_Cl] (0.10 g, 0.27 mmol) and the corresponding Ar^n^-dtpy ligand (0.27 mmol) were dissolved in toluene (60 mL), and then heated under reflux for 8 h. After this time, the reaction solution was filtered and set aside for slow solvent evaporation. The resulting red (**1**), orange (**2**–**3, 6**), and brown (**4**–**5**) crystalline solids were collected by filtration. The X-ray quality crystals of **1**–**4** were obtained directly from the mother liquor.

**1**: Yield: 77%. C_24_H_13_ClN_3_O_3_ReS_2_. Calc. C: 42.57; H: 1.94; N: 6.21. Exp. C: 42.91; H: 1.78; N: 6.16. IR (KBr, cm^−1^): 3124(w), 3045(w), 2984(w), 2865 (w) ν(Ar–H); 2019(vs), 1926(vs), 1898(vs) ν(C≡O); 1606(m), 1530(w), 1512(w) ν(C=N) and ν(C=C). ^1^H NMR (400 MHz, Acetone) δ 8.72 (s, 1H), 8.39 (d, *J* = 3.3 Hz, 1H), 8.26 (d, *J* = 3.3 Hz, 1H), 8.16 –8.11 (m, 3H), 8.11–8.05 (m, 2H), 8.02 (d, *J* = 7.3 Hz, 1H), 7.80 (d, *J* = 7.1 Hz, 1H), 7.72 –7.68 (m, 1H), 7.66–7.59 (m, 2H). ^13^C NMR not recorded due to insufficient complex solubility. DSC: I scan: T_m_ = 260 °C, II scan T_g_ = 172 °C.

**2**: Yield: 74%. C_24_H_13_ClN_3_O_3_ReS_2_ · ⅓C_7_H_8_. Calc. C: 44.68; H: 2.23; N: 5.94. Exp. C: 45.06; H: 2.16; N: 5.85. IR (KBr, cm^−1^): 3130(w), 2984(w), ν(Ar–H); 2023(vs), 1938(vs), 1880(vs) ν(C≡O); 1608(m), 1543(w) and 1503(w) ν(C=N) and ν(C=C). ^1^H NMR (400 MHz, Acetone) δ 8.95 (s, 1H), 8.75 (s, 1H), 8.39 (s, 1H), 8.37 (d, *J* = 3.3 Hz, 1H), 8.25 (d, *J* = 3.2 Hz, 1H), 8.22–8.19 (m, 1H), 8.17–8.07 (m, 4H), 8.02 (d, *J* = 7.0 Hz, 1H), 7.68–7.60 (m, 2H).^13^C NMR not recorded due to insufficient complex solubility. DSC: I scan: T_c-c_= 204 °C; T_m_ = 303 °C, II scan T_g_ = 169 °C.

**3**: Yield: 68%. C_28_H_15_ClN_3_O_3_ReS_2_ · ⅓C_7_H_8_. Calc. C: 48.07; H: 2.35; N: 5.54. Exp. C: 48.30; H: 2.38; N: 5.45. IR (KBr, cm^−1^): 3119(w), 2923(w), 2853(w) ν(Ar–H); 2023(vs), 1920(vs), 1884(vs) ν(C≡O); 1611(m) ν(C=N) and ν(C=C). ^1^H NMR (400 MHz, Acetone) δ 8.79 (s, 1H), 8.73 (s, 1H), 8.43 (d, *J* = 3.2 Hz, 1H), 8.25 (d, *J* = 3.2 Hz, 1H), 8.20 (d, *J* = 8.5 Hz, 2H), 8.13–8.09 (m, 2H), 8.06 (d, *J* = 3.0 Hz, 1H), 7.81 (d, *J* = 8.8 Hz, 1H), 7.65–7.53 (m, 4H), 7.50–7.44 (m, 1H). ^13^ C NMR (100 MHz, Acetone) δ 197.84, 195.79, 190.80, 170.92, 165.15, 156.43, 154.28, 153.04, 145.94, 144.63, 132.15, 132.12, 131.88, 131.33, 130.05, 129.84, 129.76, 129.68, 129.03, 128.38, 128.06, 127.83, 126.59, 126.53, 126.31, 126.12, 125.78, 125.28. DSC: I scan: T_m_ = 223 °C, II scan T_g_ = 185 °C.

**4**: Yield: 65%. C_28_H_15_ClN_3_O_3_ReS_2_ · C_7_H_8_. Calc. C: 51.31; H: 2.83; N: 5.13. Exp. C: 51.53; H: 3.19; N: 5.07. IR (KBr, cm^−1^): 3048(w) ν(Ar–H); 2016(vs), 1915(vs), 1890(vs) ν(C≡O); 1601(m), 1542(w) ν(C=N) and ν(C=C). ^1^H NMR (400 MHz, Acetone-d_6_) δ 9.06 (s, 1H), 9.01 (s, 1H), 8.79 (s, 1H), 8.68 (s, 1H), 8.49 (s, 2H), 8.41 (d, *J* = 3.2 Hz, 2H), 8.34 (d, *J* = 9.4Hz, 2H), 8.29 (d, *J* = 3.4 Hz, 2H), 8.23–8.12 (m, 10H), 7.62–7.59 (m, 2H). DSC: I scan: T_c-c_ = 181 °C; T_m_ = 267 °C, II scan T_g_ = 123 °C; T_m_ = 272 °C.

**5**: Yield: 72%. C_28_H_15_ClN_3_O_3_ReS_2_ · ½C_7_H_8_. Calc. C: 48.93; H: 2.48; N: 5.43. Exp. C: 49.05; H: 2.35; N: 5.33. IR (KBr, cm^−1^): 3123(w), 3057(w) ν(Ar–H); 2020(vs), 1918(vs), 1896(vs) ν(C≡O); 1606(m), 1526(w) ν(C=N) and ν(C=C). ^1^H NMR (400 MHz, Acetone) δ 8.99 (d, *J* = 8.2 Hz, 1H), 8.93 (d, *J* = 8.4 Hz, 1H), 8.81 (s, 1H), 8.41 (d, *J* = 3.3 Hz, 1H), 8.26 (d, *J* = 3.3 Hz, 1H), 8.21 (s, 1H), 8.17–8.06 (m, 4H), 8.04 (d, *J* = 8.2 Hz, 1H), 7.86–7.78 (m, 2H), 7.77–7.67 (m, 2H). ^13^C NMR not recorded due to insufficient complex solubility. DSC: I scan: T_m_ = 234 °C; T_c_ = 242 °C, T_m_ = 276 °C; II scan T_g_ = 184 °C.

**6**: Yield: 65%. C_30_H_19_ClN_3_O_3_ReS_2_. Calc. C: 47.71; H: 2.54; N: 5.56. Exp. C: 47.99; H: 2.58; N: 5.60. IR (KBr, cm^−1^): 3038(w), 2987(w), 2865(w) ν(Ar–H); 2021(vs), 1917(vs), 1891(vs) ν(C≡O); 1605(m), 1527(w) ν(C=N) and ν(C=C). ^1^H NMR (400 MHz, Acetone) δ 8.88 (s, 1H), 8.47 (d, *J* = 8.0 Hz, 1H), 8.43–8.37 (m, 3H), 8.36–8.31 (m, 4H), 8.31–8.26 (m, 3H), 8.19–8.13 (m, 2H), 8.09 (d, *J* = 3.1 Hz, 1H). ^13^C NMR not recorded due to insufficient complex solubility. DSC: I scan: T_m_ = 155 °C with decomposition.

### 3.2. Crystal Structure Determination and Refinement

X-ray diffraction data of compounds **1**–**4** were collected at room temperature on a Gemini A Ultra Oxford Diffraction with graphite monochromated MoKα radiation (λ = 0.71073 Å). Diffraction data collection, cell refinement, and data reduction were performed using the CrysAlis^Pro^ software [87]. The structures were solved via the direct methods using SHELXS and refined by full-matrix least-squares on *F*^2^ using SHELXL-2014 [88,89]. The Olex2 program was used for all calculations [90]. All non-hydrogen atoms were refined anisotropically. All hydrogen atoms were positioned in geometrically idealized positions and were allowed to ride on their parent atoms with *d*(C–H) = 0.93 Å, *U*_iso_(H) = 1.2 *U*_eq_(C) (for aromatic) and *d*(C–H) = 0.96 Å, *U*_iso_(H) = 1.5 *U*_eq_(C) (for methyl). The methyl groups were allowed to rotate about their local threefold axis. Details of the crystallographic data collection, structural determination, and refinement for **1**–**4** are listed in Appendix A, ESI†, whereas the selected bond lengths and their angles are listed in Appendix A, ESI†.

### 3.3. Physical Measurements

Elemental analyses (C, H, N) were conducted on a Perkin–Elmer CHN–2400 analyzer. The FT-IR spectra were carried out using KBr pellet technique with Nicolet iS5 FT-IR spectrophotometer (4000–400 cm^–1^). The ^1^H NMR and ^13^C NMR spectra were collected (295 K) on Bruker Avance 400 NMR spectrometer.

Electronic absorption measurements were performed using ThermoScientific Evolution 220 UV/Vis Spectrometer (in case of solutions) and/or Jasco V570 UV–Vis–NIR Spectrometer (in case of films deposited on a glass substrate and as blends with poly(N-vinylcarbazole) (PVK): 2-(4-tert-butylphenyl)-5-(4-biphenylyl)-1,3,4-oxadiazole (PBD) on a glass substrate).

Electrochemical tests were studied with the use of Eco Chemie Autolab PGSTAT128n potentiostat (argon-saturated dichloromethane, c = 10^−3^ mol/L, 0.1M Bu_4_NPF_6_ as a supporting electrolyte). Glassy carbon electrode (⌀2 mm), platinum coil, and silver wire were used as working, auxiliary, and reference electrodes, respectively. Cyclic voltammetry (CV) and differential pulse voltammetry (DPV) were recorded with scan rates 0.1 V/s and 0.01 V/s, respectively. All the results were calibrated on ferrocene (Fc) as internal standard.

Steady-state photoluminescence (PL) spectra were measured with an FLS-980 fluorescence spectrophotometer in argon-bubbled chloroform solution at room temperature and in an ethanol:methanol (4:1 v:v) rigid matrix at 77K. Quantum yields were tested using an integrating sphere with the solvent (CHCl_3_) as a blank. The compounds were excited (**1**–**6, respectively**) with wavelengths equal to 463 nm, 487 nm, 475 nm, 475 nm, 500 nm, 411 nm, which correspond to the excitation range yielding a single phosphorescence band disturbed neither by fluorescence signal nor reabsorption of the sample. All the emission spectra were corrected in terms of the sensitivity of the monochromator, detector, sphere coating, and optics to wavelength detected. Time-correlated single-photon counting (TCSPC), or multi-channel scaling (MCS) methods were used for measurement of photoluminescence lifetime. Set of picosecond pulsed diode lasers (Edinburgh Instruments) and 60W microsecond Xe flash lamp were utilized for TCSPC and MCS, respectively. Additionally, IRF was measured for the analysis of fluorescence decays. Photoluminescence spectra in solid state as film deposited on a glass substrate and as blends with poly(N-vinylcarbazole) (PVK): 2-(4-tert-butylphenyl)-5-(4-biphenylyl)-1,3,4-oxadiazole (PBD) on a glass substrate were collected on a Hitachi F-2500 spectrometer.

A precise voltage supply (Gw Instek PSP-405) with the sample fixed to an XYZ stage was applied to collect electroluminescence (EL) spectra, and all measurements were performed using the procedure reported in our previous work [55].

Differential Scanning Calorimetry studies were carried out with TA-DSC 2010 apparatus under nitrogen atmosphere, with heating rate 20 °C/min. ATLAS 0531 Electrochemical Unit and Impedance Analyzer potentiostat was used to perform electrochemical measurements [55]. Thickness of active layers was determined using an atomic force microscope (AFM) Topometrix Explorer TMX 2000.

### 3.4. Computational Details

Calculations (geometry optimization, absorption, and emission spectra) were performed using the GAUSSIAN-09 program package [91] at the DFT level with the PBE1PBE [92,93] hybrid exchange–correlation functional, and the def2-TZVPD basis set for rhenium and def2-TZVP basis set for other elements [94,95,96]. Vibrational frequencies were calculated on the basis of the optimized geometry to verify that each of the geometries corresponds to a minimum on the potential energy surface. In calculations, the solvent environment (chloroform) was accounted for by using polarizable continuum model (PCM) [97,98,99]. Absorption properties were calculated via the TD-DFT method on the basis of optimized ground-state geometries.

### 3.5. Femtosecond Transient Absorption Experiment

The fsTA spectra were measured using a pump–probe transient absorption spectroscopy system (Ultrafast Systems, Helios) described elsewhere [55], and including a regenerative 1 kHz amplified femtosecond Ti:sapphire laser system (Astrella, Coherent) to generate 800 nm pulses (100 fs duration, 5 mJ pulse energy) in addition to an optical parametric amplifier (Light Conversion, TOPAS prime) to obtain 355 nm pump pulses. A mechanical chopper synchronised to one-half of the laser repetition rate was used for detecting series of spectra pumped and unpumped. The pump beam was then depolarised to mimic the dynamical orientation changes of the molecules. CaF_2_ crystal was used for generation of a white light continuum probe. The delay time between the pump and probe pulses was controlled using a delay line over a time scale up to 7.6 ns. A CCD detector was used for the detection of transient absorption signals. Samples in chloroform solution (c–125 μM; A_355_: 0.4–0.75) were placed in 2 mm quartz cuvettes and stirred to prevent sample damage during the experiment. Prior to the femtosecond analysis, fluence dependence and photodamage tests were conducted in order to set the fsTA experiment conditions. Surface Xplorer (Ultrafast Systems) and Optimus^TM^ software [100] were employed for the processing of transient absorption data (background correction, scattered light subtraction, solvent signal contribution subtraction, and spike removal) and analysis. The coherent artefact analysis provided information about instrument response function (FWHM of ~170 fs), light chirp and group velocity dispersion. The linear unidirectional sequential model [100] was implemented for global analysis of the TA maps. Results from the analysis show that deconvolution of the transient spectra into evolution-associated spectra (EAS) and decay-associated spectra (DAS) was obtained.

## 4. Conclusions

In summary, a series of Re(I) carbonyl complexes based on aryl-substituted 2,6-di(thiazol-2-yl)pyridines (Ar-dtpy) were synthesized and thoroughly investigated via cyclic voltammetry, absorption and emission spectroscopies, transient absorption, in conjunction with the density functional theory (DFT), and time-dependent DFT methods in order to determine the effect of a number of condensed aromatic rings and linking mode of the aryl substituent on the thermal, electrochemical and spectroscopic properties of [ReCl(CO)_3_(Ar-dtpy-κ^2^N)]. The replacement of 1-naphthyl-, 2-naphthyl-, 9-phenanthrenyl groups with 9-anthryl, 2-anthryl or 1-pyrenyl groups resulted in strong modification of the nature of the lowest triplet excited-state from ^3^MLCT to ^3^IL_anthracene_ (in **3**) or ^3^IL/^3^ILCT ((Aryl)/(Aryl→dtpy) (in **4** and **6**). The 1-pyrenyl-appended Re(I) complex (**6**) displayed an increased lifetime in solution at room temperature (1.20 μs) in comparison to [ReCl(CO)_3_(Ar-dtpy-κ^2^N)] with 1-naphthyl-, 2-naphthyl-, 9-phenanthrenyl groups (~6ns), which was rationalized by the formation of a triplet-state equilibrium between ^3^MLCT and ^3^IL/^3^ILCT. Based on the fsTA results of **6**, the following deactivation model was proposed ^1^IL_pyrene_/^1^ILCT_pyrene__→__dtpy_ → ^1^MLCT → ^3^MLCT→ ^3^IL_pyrene_/^3^ILCT_pyrene__→__dtpy_. The photophysical properties of isomeric complexes [ReCl(CO)_3_(9-anthryl-dtpy-κ^2^N)] and [ReCl(CO)_3_(2-anthryl-dtpy-κ^2^N)] were found to be strongly dependent on the anthryl linking mode and excitation wavelengths. Most notably, the Re(I) complex with 2-anthryl group excited at 475 nm displayed phosphorescence contributed to by emissions from ^3^MLCT and ^3^IL_An_/^3^ILCT_An__→_
_dtpy_. Femtosecond transient absorption spectroscopy studies of these systems confirmed that less steric hindrance between these chromophores in [ReCl(CO)_3_(2-anthryl-dtpy-κ^2^N)] facilitates the formation of the ligand-centred triplet excited state, and is responsible for larger contribution of the ^3^ILCT_anthracene__→__dtpy_ excited state compared to the isomeric complex [ReCl(CO)_3_(9-anthryl-dtpy-κ^2^N)].

## Data Availability

Crystallographic data for **1**–**4** were deposited into the Cambridge Crystallographic Data Centre, CCDC 2195019–2195022. Copies of this information may be obtained free of charge from the Director, CCDC, 12 Union Road, Cambridge CB2 1EZ, UK; deposit@ccdc.cam.ac.uk or www.ccdc.cam.ac.uk; Fax: +44-12-2333-6033.

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
