# Peer review of "Controlling of Photophysical Behavior of Rhenium(I) Complexes with 2,6-Di(thiazol-2-yl)pyridine-Based Ligands by Pendant π-Conjugated Aryl Groups"

_ijms, 2022, doi:10.3390/ijms231911019_

Round 1

Reviewer 1 Report

In the manuscript, the authors prepared a series of Re(I) carbonyl complexes with aryl-substituted 2,6-di(thiazol-2-yl)pyridines, and investigated their ground and excited state properties through a wide range of characterization techniques, including CV, UV-vis and PLE spectroscopies, transient absorption, as well as the DFT and TD-DFT methods. Hence, the work is well organized, the analysis is fine and reasonable, and the manuscript is well prepared, I can recommend acceptance following some minor modifications,

1. The style of some figures is not consistent, including the style of unit and coordinate lables in Fig 3 in main text. The x-coordinate in IR spectra should be corrected as 'wavenumber' in SI materials.

2. Throughout the ms as well as figures, the authors defined lamda(PE) as excitatation wavelength, however, in most case, lamada(ex) or lamda(PLE) should be used instead.

3.  From line 214, the authors comment on the absence of superimposition or lack of characteristic vibronic progression for the anthracene group. The similar absorbance or PL spectra could be even observed in anthracene-linker MOF solid, naphthol-containing metal-organic rhomboid (see Inorg. Chem. 201 8, 57, 1448914492) and etc.

4. The fs-TA analysis of excited state dynamics in metal complexes in some recent literatures were also reported, such as Chem. Commun., 2020, 56, 12057-12060

Author Response

  1. The style of some figures is not consistent, including the style of unit and coordinate lables in Fig 3 in main text. The x-coordinate in IR spectra should be corrected as 'wavenumber' in SI materials.

Authors’ reply: It has been done.

  1. Throughout the ms as well as figures, the authors defined lamda(PE) as excitatation wavelength, however, in most case, lamada(ex) or lamda(PLE) should be used instead.

Authors’ reply: It has been done.

  1. From line 214, the authors comment on the absence of superimposition or lack of characteristic vibronic progression for the anthracene group. The similar absorbance or PL spectra could be even observed in anthracene-linker MOF solid, naphthol-containing metal-organic rhomboid (see Inorg. Chem. 201 8, 57, 14489−14492) and etc.

Authors’ reply: The reference suggested by reviewer has been added.

  1. The fs-TA analysis of excited state dynamics in metal complexes in some recent literatures were also reported, such as Chem. Commun., 2020, 56, 12057-12060.

Authors’ reply: The reference suggested by reviewer has been added.

Reviewer 2 Report

Comments:

This paper for “Controlling of photophysical behavior of rhenium(I) complexes 2 with 2,6-di(thiazol-2-yl)pyridine-based ligands by pendant 3 π-conjugated aryl groups” was presented. The manuscript and experimental results are good in the average. The authors logically used their ideas to create their target compounds. It is full to perform the complete analysis reports in this study. The scientific contents were well presentation and gave the contribution to chemistry. I recommend the publication of this manuscript only after a minor revision. Please find the comments below:

1.         The authors have conducted related research in previous studies, such refs 26, 27 and 28. Could the authors address some comments or discussions on these studies? For example, why do the authors always use the symmetrical ligands with different backbones? The key is the pendant π-conjugated aryl groups or the “dtpy” ligands. In this study, one side of the “dtpy” ligand didn’t coordinate with the metal center. These results were in agreed with the authors of previous studies. Does it possibly have some functionality of the other side “dtpy” ligand?

2.         Why did the hydrogen values have three decimal places in the elemental analyses of these complexes?

Author Response

  1. The authors have conducted related research in previous studies, such refs 26, 27 and 28. Could the authors address some comments or discussions on these studies? For example, why do the authors always use the symmetrical ligands with different backbones? The key is the pendant π-conjugated aryl groups or the “dtpy” ligands. In this study, one side of the “dtpy” ligand didn’t coordinate with the metal center. These results were in agreed with the authors of previous studies. Does it possibly have some functionality of the other side “dtpy” ligand?

Authors’ reply: The introduction has been modified, and the choice of the 2,6-di(thiazol-2-yl)pyridines functionalized with π-extended polycyclic aromatic hydrocarbons for designing new Re(I) chromophores has been explained in more detail. Formation of Re(I) carbonyls with terpy-like ligands coordinated in tridentate way is possible, but it requires a totally different synthetic methodology, as reported in  Inorg. Chem. 2021, 60, 7079 and Inorg. Chem. 2021, 60, 13251333.

  1. Why did the hydrogen values have three decimal places in the elemental analyses of these complexes?

Authors’ reply: It has been changed.